# Simulation of slip transients and earthquakes in finite thickness shear zones with a plastic formulation

Xinyue Tong [1] & Luc L. Lavier [1]

We perform numerical experiments of damped quasi-dynamic fault slip that include a rate-and-state behavior at steady state to simulate earthquakes and a plastic rheology to model permanent strain. The model shear zone has a finite width which represents a natural fault zone. Here we reproduce fast and slow events that follow theoretical and observational scaling relationships for earthquakes and slow slip events (SSEs). We show that the transition between fast and slow slip occurs when the friction drop in the shear zone is equal to a critical value, $\Delta\mu_c$. With lower friction drops, SSEs use nearly all of mechanical work to accumulate inelastic strain, while with higher friction drops fast slips use some of the mechanical work to slip frictionally. Our new formulation replaces the state evolution of rate and state by the stress evolution concurrent with accumulation of permanent damage in and around a fault zone.

[1] Department of Geological Sciences and Institute for Geophysics, Jackson School of Geosciences, The University of Texas at Austin, Austin TX 78712, USA. Correspondence and requests for materials should be addressed to X.T. (email: dennistong@utexas.edu)

During an earthquake, elastic strain energy is released as energy carried by seismic waves, kinetic energy released as slip along a frictional interface, energy needed to form new fractures, and thermal energy[1]. For the last two decades, slow slip events (SSEs) have been observed at many subduction zones[2]. Like megathrust earthquakes, SSEs represents shear slip at the plate interface[3]. Unlike fast earthquakes, SSEs are mostly observed through long-term geodetic observations[4,5]. They are also part of the total energy released through the seismic cycle and since they occur in areas adjacent to the source of very large earthquakes they could potentially load the seismogenic locked patch[3,6–8]. SSEs release tectonic stress at lower-frequency than those released by fast earthquakes[6,8,9], likely due to the lower slip and rupture propagation velocities of SSEs[3,6,8,10,11]. These observations suggest that the elastic energy loaded inter-seismically is released in fundamentally different ways during SSEs and fast earthquakes. To understand the different slip behaviors of a fault over secular time scales and the interaction between long-term tectonic (LTT) and seismic processes we first need to understand the differences in energy partitioning between fast earthquakes and SSEs. Here we present a first-order attempt at quantifying the partitioning between kinetic and strain energy that are respectively associated with the release of elastic energy in slip and in permanent deformation on fault zones.

There are many numerical studies of the conditions for the emergence of earthquakes and SSEs based on the rate-and-state dependent friction including phenomena such as heat pressurization or dehydration reactions[12–19]. Other attempts model slow slip in viscoelastic materials[20–24], more adequate for the pressure and temperature conditions under which deep SSEs are observed[6,25–30]. Finally a few attempts at modeling the earthquake cycle in models of LTT deformation use rate and state dependence in viscoplastic formulation of shear zones[31–33]. The rate and state approach often uses the aging law[16,17] to describe the time-dependent stickiness or stiffness of two frictional surfaces in contact, allowing for the numerical simulation of earthquakes on predefined fault surfaces. However, the physical meaning of the aging parameter in a shear zone of finite thickness rupturing over multiple surfaces is not fully understood[34–38].

Here we develop an approach that considers that aging is the result of the damage history in the fault zone[35,39,40]. When the shear zone yields, the incremental accumulation of plastic strain/damage results in incremental changes in shear stress. The stress variations are functions of dynamic friction, $\mu_d$ that is imposed by the rate-dependent friction law at steady state[12–14,16,17,41,42]:

$$\mu_d = \left( \mu_0 + (a - b) \ln \frac{V}{V_0} \right) \quad (1)$$

where $V$ is velocity magnitude, $\mu_0$ is reference friction coefficient, and $V_0$ is reference slip velocity magnitude. $a$ and $b$ are dimensionless frictional stability parameters. $\mu_d$ increases with increasing $V$ (rate strengthening) if $a > b$, and $\mu_d$ decreases with increasing $V$ (rate weakening) if $a > b$. Our model shows that the transition between fast and slow transients occurs when $a - b = -0.0006$.

## Results
### Reference model.
To study how the properties of our formulation compare with those of natural slip events, we first test it in an experimental setup similar to a single spring slider system at tectonic scale (Fig. 1a). The 10 km thick upper layer (overriding plate) is elastic. The 2 km thick bottom layer represents the shear zone (where localized fault zones form, e.g., subduction interface). Materials in both layers have same density, bulk, and shear modulus, and cohesion (The values of input parameters are listed

in Supplementary Table 1). Fault slip is driven at a boundary velocity of 32 mm yr$^{-1}$ applied at the top of the model. The sides are free and the bottom is fixed. For simplicity we do not impose any geometrical or rheological heterogeneities in the fault zone (Fig. 1a). The material inside shear zone is uniformly velocity-weakening ($b - a > 0$). Our experimental parameter space includes the fault length, $L$, and the value of $b - a$. We vary $L$ from 20 to 90 km. For each fault length, we run experiments with a different value of $b - a$ varying from 0.004 to 0. All numerical experiment last 10,000 years.

Figure 1b shows examples of simulated slip events for $b - a = 0.004$. They are measured at the fault interface (Fig. 1a). The minimum and maximum slip rates vary between $10^{-10}$ and $10^{-1}$ m·s$^{-1}$. The latter similar to earthquake slip rates and the former lower than the background loading rate. After a short period of stress increase, elastic loading of the upper plate and plastic strain accumulation at the interface between the upper plate and the shear zone form a fault zone. A discussion in the supplementary materials discusses the fault zone evolution dependence on the mesh resolution. Long-term creep and strain accumulation occur when the shear stress exceeds yield stress defined by the reference coefficient of friction and cohesion[43]. The fault zone creeps inter-seismically until a critical shear stress for unstable slip is reached. The interseismic period is ~320 years. For one event (Fig. 1c), we find that the slip, stress drop, and event are similar to that of an earthquake[44–46].

### Velocity stepping test.
To compare our results with conventional rate-and-state laboratory experiments, we perform a tectonic scale velocity stepping test using our model setup[37,47–49]. We first apply a loading rate of $10^{-9}$ m s$^{-1}$ for 3000 years. We then abruptly change the rate to $10^{-2}$ m vs$^{-1}$ for about 66 h and drop it back to $10^{-9}$ m s$^{-1}$. We plot the friction coefficient, maximum shear stress, and slip velocity in log scale over displacement during the numerical experiment (Fig. 2a). We remark that the dynamic friction coefficient in the fault zone changes instantaneously when the velocity abruptly increases to $10^{-2}$ m s$^{-1}$. However, the stress change is delayed when compared to the change in slip rate. We also plot the resulting effective friction coefficient (Fig. 2b). This effective or equivalent resulting friction coefficient is calculated the yield stress ($\tau_{max} \cos \phi_d$) divided by normal stress. Finally, we zoom-in the beginning and ending portions of the period of fast loading. We find that changes in friction coefficient follow a pattern equivalent to the resulting effect of the rate and state aging law (Fig. 2b, c). Although we only consider rate-dependent friction in this implementation, the modeled frictional behavior registers both a direct and an evolving effect which are comparable to that of the rate-and-state friction law. In our formulation, the variations in the state of stress (equations 6–11) depend on the deformation history through plasticity[35,39] (Supplementary Figure 2) and replace the frictional evolution imposed by the aging law[15–17,50].

### Emergence of fast slips and slow transients.
In order to test the properties of our elasto-plastic rate and state formulation, we vary both the length and the $b - a$ value of the shear zone. As $b - a$ decreases from 0.004 to 0, slip behavior changes from fast slip transients similar to earthquakes to small creep events (Fig. 3a). Slip instability occurs when the magnitude of the frictional resistive force per unit area, $F$ decreases faster than the stiffness, $\kappa$, of the material surrounding the fault as a function of characteristic slip distance, $\mathcal{L}$[51]. This phenomenon is independent of the evolution law used[41]. This frictional resistive force can be expressed as, $\Delta F = \Delta \mu \sigma_n$ with the change in friction, $\Delta \mu = b - a$[16,17], and $\sigma_n$ the normal stress. In our experiments

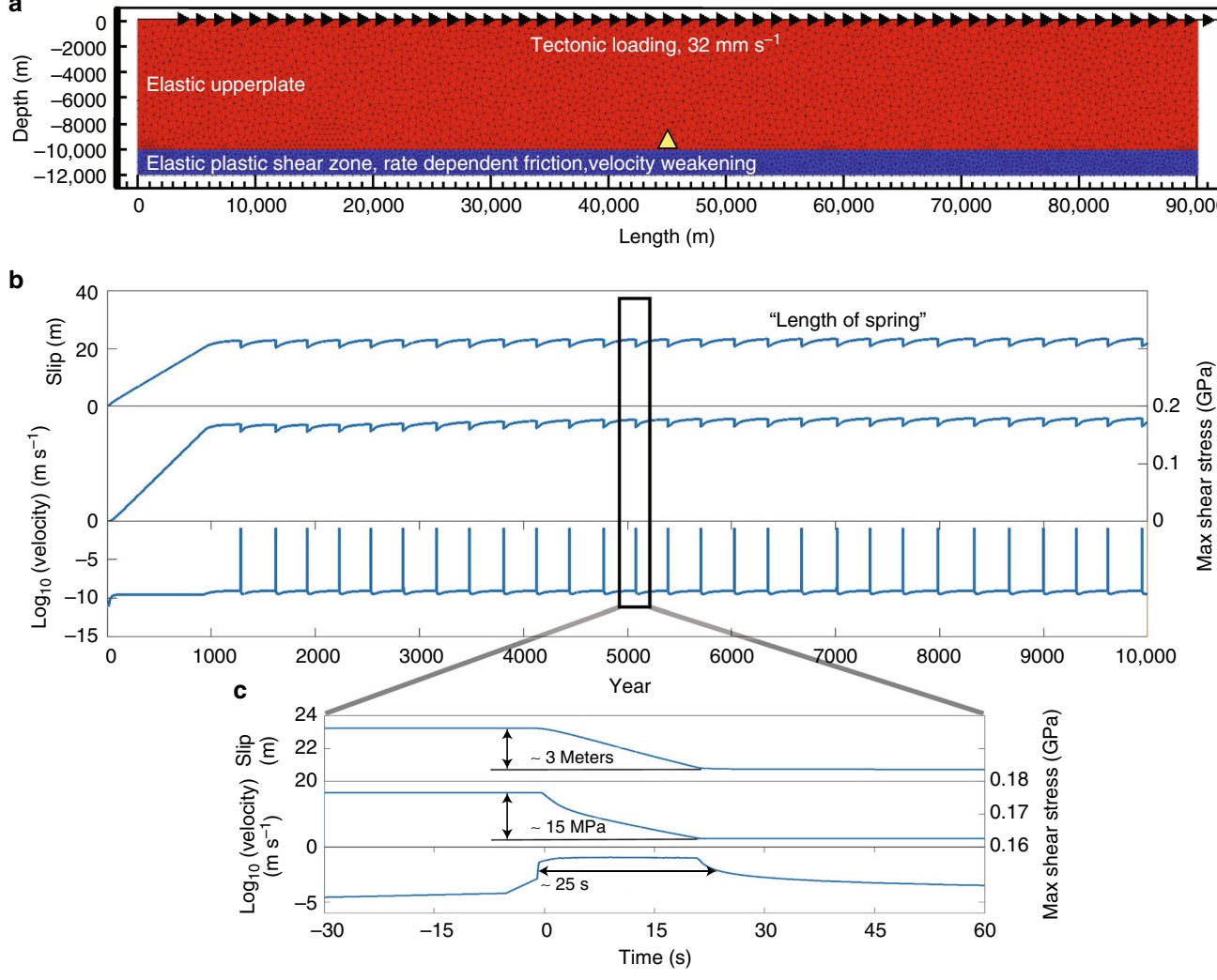

**Fig. 1** A single spring slider experiment in tectonic scale. **a** Model setup for a 90 km fault length. $a-b = -0.004$. Yellow triangle marks the element where the measurements are recorded. **b** Slip, maximum shear stress, and velocity are plotted over time. We stop the simulation at 10,000 year. **c** Zoom-in. The event has about 3 meter of co-seismic slip and 15 MPa of stress drop. The peak slip velocity reaches to 0.1 m s$^{-1}$, and it lasts for about 25 s

slow slip transients emerge at a critical value of $\Delta\mu_c = 0.0006$ for all fault length (Fig. 3b). Fault zones with $\Delta\mu > \Delta\mu_c$ have an earthquake like behavior and transition to creep events when $\Delta\mu < \Delta\mu_c$. For two or three-dimensional fault zones, $\kappa$ scales inversely with $L$, $\kappa = \eta G/L$, where $\eta$ is a geometrical constant in the order of unity[52] and $G$ is the shear modulus. Therefore, for $L = 60$ km, the stiffness in our model is $\kappa = 1.1 \cdot 10^6$ Pa m$^{-1}$, and the characteristic slip distance, $\mathcal{L}$ is ~0.15 m. $\mathcal{L}$ is estimated using the relationship, $\sigma_c = \kappa\mathcal{L}/\Delta\mu_c$[52], with $\sigma_c = \sigma_n = \rho g h = 274$ MPa. Previous studies[15] show that $\gamma = L/h^*$ with $h^* = 2G\mathcal{L}/(\pi(1-v)(b-a)\sigma_n)$ is controlling the slip behaviors in the fault zone, which $v$ is the Poisson's ratio. In our experiments this ratio is also equal to $\gamma = \Delta F/\mathcal{L}\kappa$. When $\gamma < 1$ the force drop over $\mathcal{L}$ is greater than $\kappa$ and the fault zone cannot deform enough to accommodate the stress drop, it therefore slips frictionally in an earthquake, as well as accumulate deformation across the fault zone. If $\gamma < 1$ the force drop over $\mathcal{L}$ is smaller than $\kappa$ and the fault zone deformation is enough to accommodate the stress drop. As a result, it mostly accumulates deformation across the fault zone and creeps.

**Scaling relationships.** To compare the characteristics of our simulated events to both theoretical and observed scaling relationships, we measured accumulated slip, maximum shear stress,

and slip velocity for all simulated events. We identify fast and slow transient events based on their maximum slip velocity magnitude, $V_{max}$. Slow transient events have a maximum slip velocity more than one order of magnitude larger than the background tectonic velocity, and the $V_{max}$ for earthquakes is of the order of 0.1 m s$^{-1}$ (Fig. 3). We use a plane strain formulation, which means the fault zones are infinitely long in the strike direction. We define the equivalent moment as $M_o = \bar{D}GL \times 4L$. For each event, we measure the duration of slip and use it as the characteristic time to evaluate the scaling relationships.

For slip on a rectangular fault, the stress drop ($\Delta\sigma$) scales linearly with strain change, $\Delta\sigma = CG(\bar{D}/L)$, where $C \approx 1$[44]. We plot the strain drop of the simulated events versus maximum shear stress drop (Fig. 4a). The size of the diamonds represents the length of the modeled fault, $L$ and the color scale shows the value of $a-b$. Filled and unfilled diamonds represent fast and slow transients respectively. Our results show that the stress drop has a dependence on $a-b$ which follows the prediction of the rate and state-dependent friction studies[53]. The stress drop for slow transients is about one order of magnitude smaller than the stress drop for fast earthquakes. Statistically, the results show that the stress drop and strain change scales linearly. Our result also show that stress drop is independent of fault size, which is in good agreement with observations[44]. We also plot co-seismic slips

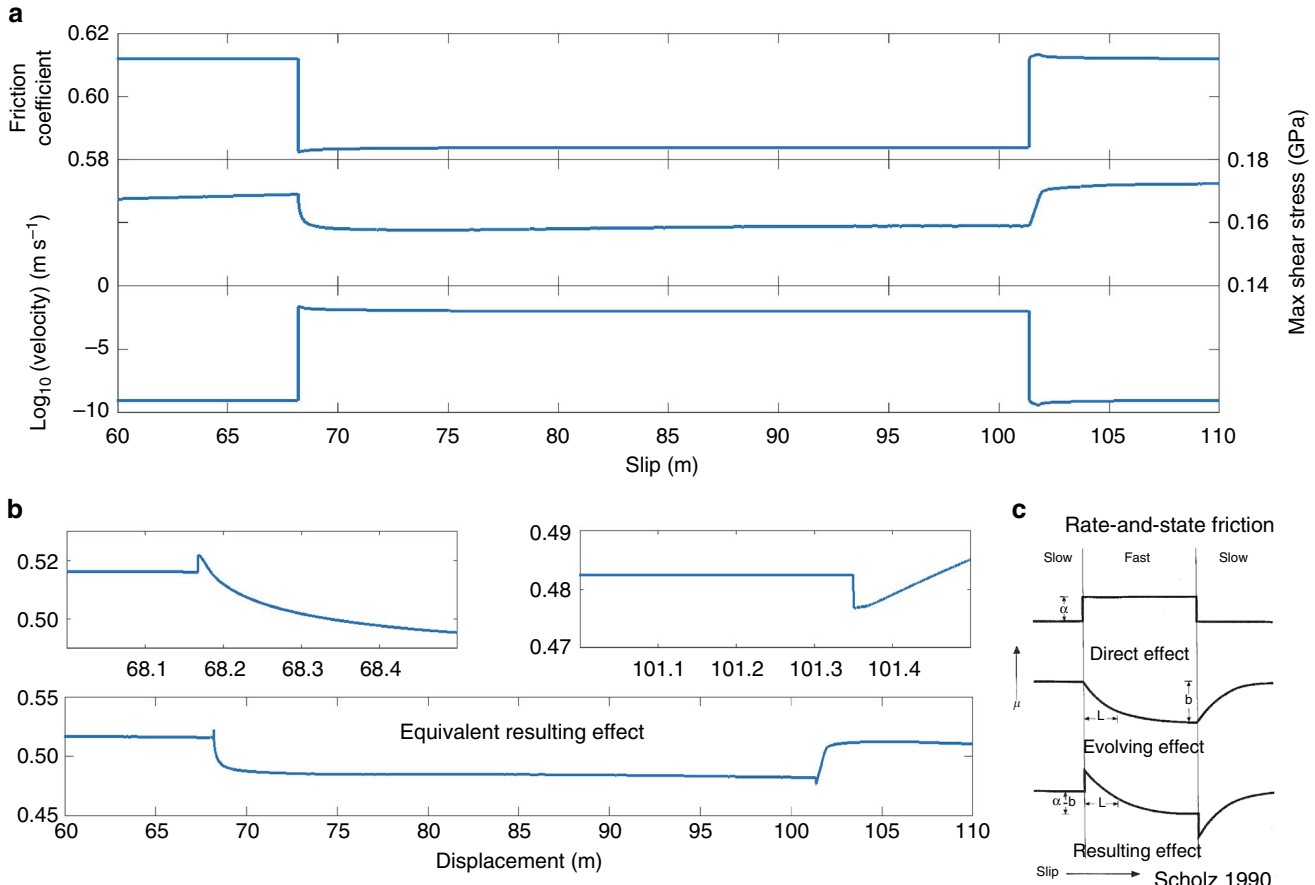

**Fig. 2** Velocity stepping test. **a** Friction coefficient, maximum shear stress, and velocity against displacement. Rate-dependent friction shows the direct response to the velocity change. The maximum shear stress also changes but with delay. **b** Calculated equivalent resulting effects. The resulting effect has an instantaneous increase followed by a gradual decrease. **c** Direct, evolving, and resulting frictional effect of the rate-and-state-dependent friction law

against durations (Fig. 4b). For earthquakes, the duration $T$ needed for slip to reach its maximum value at any point along the fault is predicted to be $T = G \cdot \bar{D}/(V_{shear} \cdot \Delta\sigma)$[46], where $V_{shear}$ is the shear wave speed. The simulated earthquakes follow the linear relationship between duration and co-seismic slip. However, slow events depart substantially from that scaling as $\bar{D} \sim T^{1/15}$.

Finally, we plot $T$ for all fast and slow transients versus $M_o$ (Fig. 5). Ide, et al.[3] proposed scaling relationships for both fast and slow earthquakes. For slow earthquakes, $M_o$ should scale nearly linearly with $T$, while for fast earthquakes, $M_o$ scales with $T^3$. The simulated fast slips are following the proposed fast earthquakes scaling. Slow transients are scattered over very variable durations from hours to years, but most plot near or in the proposed slow transients scaling. We also compare the results with observations complied by Peng and Gomberg[11]. The duration versus moment distribution of the numerical results from our study and the observations of subduction zone earthquakes are similar for both fast and slow events.

## Discussion
Our results show that SSEs emerge as $a-b$ tends to 0 and that this process is independent of fault size. Slip events simulated by our model follow theoretical scalings[44,46] and are in general agreement with observations[3,11,44]. These scaling relationships emerge from a simplified model setup that does not include any imposed spatial heterogeneities. Our results show that earthquakes and slow transients have different characteristics. In order to understand the processes controlling these differences, we analyze the results from the perspective of the energy spent slipping (kinetic)

and that used to damage the shear zone (strain). We calculate the released kinetic, $KE = \rho V^2/2$, and strain energy, $U = \Delta\left[\sigma_1^2 + \sigma_2^2 + \sigma_3^2 - 2\nu[\sigma_1\sigma_2 + \sigma_2\sigma_3 + \sigma_1\sigma_3]\right]/2E$ per unit volume and define the work efficiency:

$$W_{eff} = \frac{U}{KE + U}. \qquad (2)$$

$\rho$, $E$, and $\nu$ are material's density, Young's modulus, and Poisson's ratio and $\sigma_{1,2,3}$ are the principal stresses. For fast transients, $W_{eff}$ decreases to 65% as the fault experiences many episodes of inelastic strain accumulation until $W_{eff}$ abruptly reaches 0% when all the energy is released as slip. For slow transients, $W_{eff}$ is always close to 100% as most of the elastic energy is released in inelastic strain accumulation. Therefore, earthquakes and SSEs differ in the way they partition energy (Fig. 6). Given the transition between fast and slow transients occurred for $\Delta\mu_c = 0.0006$ for any fault length, $L$, in our model setup (Fig. 3). If $\Delta\mu > 0.0006$, $W_{eff}$ oscillates between ~98 and 65%, which for the fast slip the fraction of kinetic energy increases to 35% in bursts (Fig. 6a). After the earthquake is only kinetic as slip velocity decreases to tectonic velocity. If $\Delta\mu < 0.0006$, most of the work done is damaging the fault zone. The incremental nature of the change in $W_{eff}$ is related to the fact that the Mohr Coulomb formulation decreases friction and accumulate strain in increments. Before any event the fault is yielding and creeps plastically. The observed transition in energy partitioning coincides with $\gamma = 1$. For an earthquake $\gamma > 1$ and the energy exceeding the deformation potential of the fault zone is used to accumulate frictional slip (Kinetic Energy) and is likely dissipated as heat in a

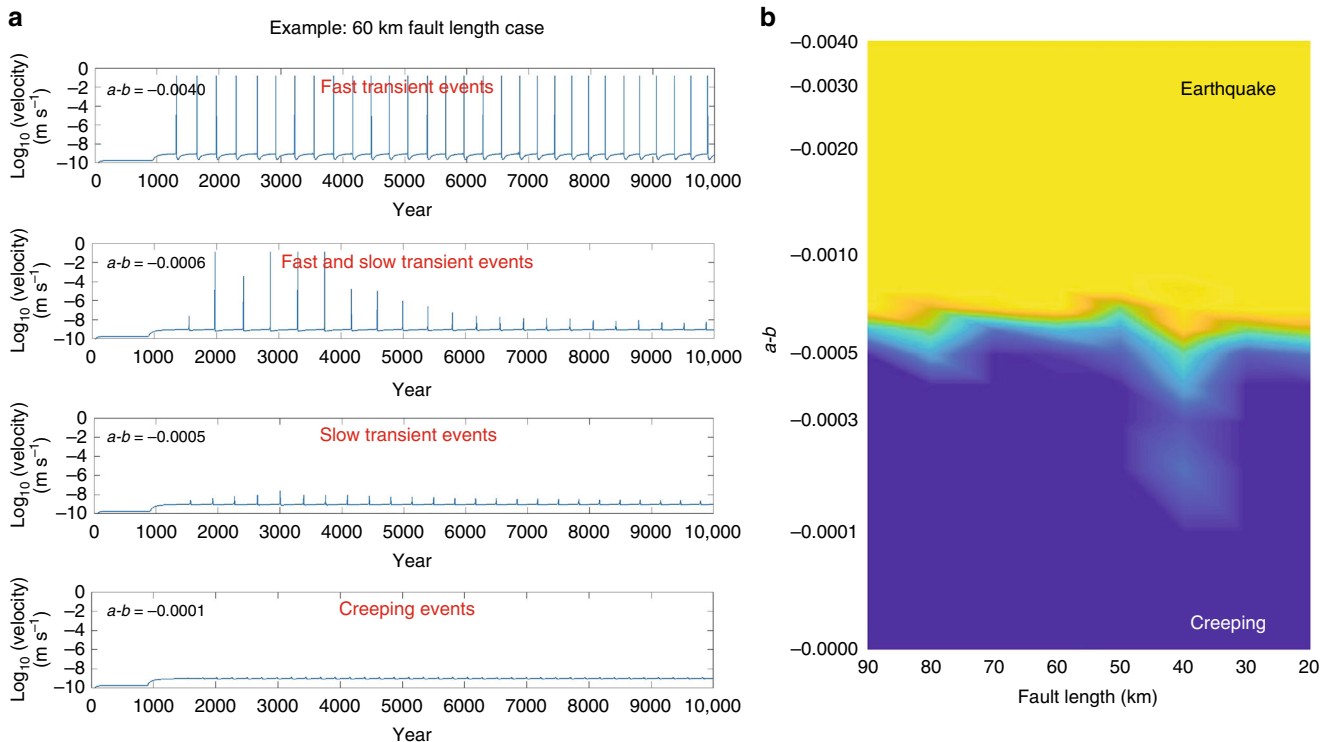

**Fig. 3** Emergence of fast slips and slow transients. **a** Slip mode transition on models with 60 km fault length and different $a-b$ values. **b** Slip behavior for different fault lengths and $a-b$ values. The scale bar shows the ratio between number of earthquakes and slow transient events. Yellow: the model only generates earthquakes. Blue: the model only generates creeping events

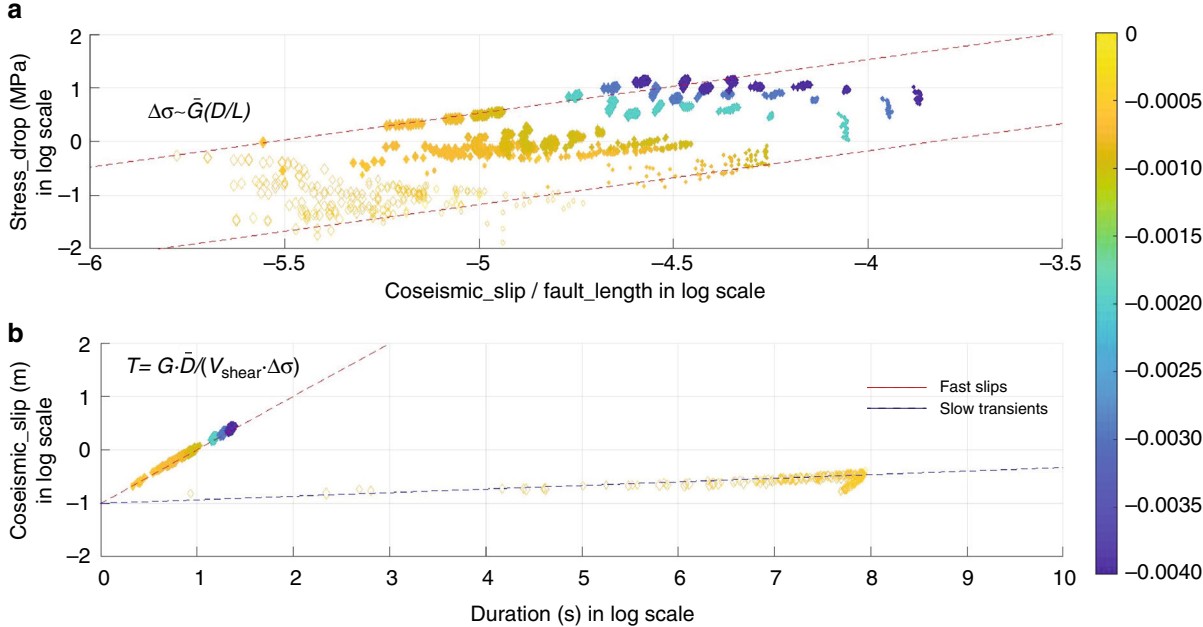

**Fig. 4** Seismic scaling relationships. **a** Relationship between strain change and stress drop. For fast and slow transient events strain drop generally scale linearly (dashed lines) with stress drop. Observations show that stress drop is independent of fault size. Within the linear dependence our results agree with observations. **b** Scaling relationship between co-seismic slip and duration. Duration of simulated earthquakes scales linearly with co-seismic slip. However, slow transient departs widely from this scaling. Filled diamonds: earthquake events. Unfilled diamonds: slow transients. The size of the diamonds represents the length of the modeled fault, $L$ and the color scale shows the value of $a-b$, from 0 (yellow) to $-0.004$ (blue)

natural system[54]. For SSEs, $\gamma > 1$ and most of the elastic energy available is used to deform the fault zone plastically and creep.

Our results therefore depart from the existing rate-and-state numerical studies that assign the emergence of slow earthquakes to processes by reducing the ratio of fault length to nucleation length[15]. We now use the equivalent ratio $\gamma$ and assign a physical meaning to the emergence of SSEs or earthquakes that corresponds to the energy partitioning between frictional slip and damage in a fault zone of finite thickness. Our formulation in DynEarthSol3D (DES3D) is also highly flexible and can be used

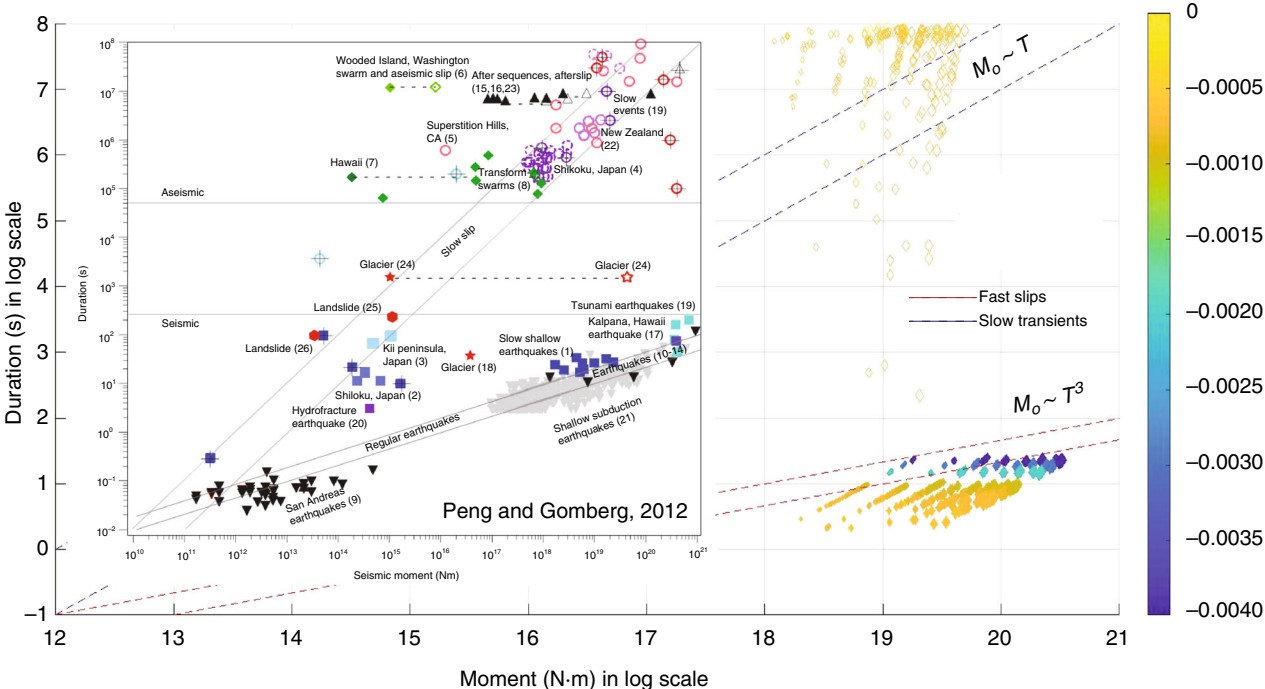

**Fig. 5** Scaling relationship between duration and moment. Dashed lines represent the proposed scalings. The simulated fast events plot in the region for fast earthquakes. Slow transients are scattered in a large space, but most of them are in or above the slow earthquakes region. Filled diamonds: earthquake events. Unfilled diamonds: slow transients. The size of the diamonds represents the length of the modeled fault, $L$ and the color scale shows the value of $a-b$, from 0 (yellow) to $-0.004$ (blue)

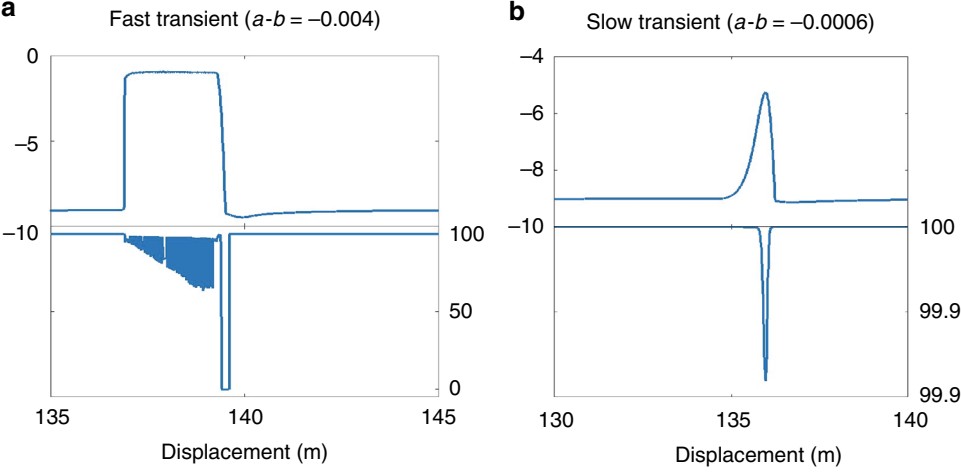

**Fig. 6** Work efficiency. Slip velocity and work efficiency for both simulated **a** fast and **b** slow transient events

to study the emergence of SSEs and megathrust earthquakes at rough or smooth subduction interfaces of finite thickness under varying pore fluid pressure conditions. For example, subduction zones with weak fault zones are thought to be the locus of large earthquakes while strong and rough interfaces are thought to be the locus of smaller events (i.e., Southern Chile[55] vs Hikurangi[56]). Our experiments and the slip instability condition[51] suggest that fault zones composed of weak material with low shear modulus could indeed the locus of large earthquakes and strong or rough material can lead to the emergence of slow earthquakes. However, if $b-a$ is close to zero and smaller than $\Delta\mu_c$ even weak sediments with low fluid pore pressure will always lead to slow transients or creep events. On the other hand, if $b-a$ is larger than $\Delta\mu_c$,

megathrusts earthquakes may occur with a rough and strong basaltic oceanic crust under high pore fluid pressure.

## Methods

**DES3D and fast lagrangian analysis of continua (FLAC).** At present, the most used formulation of rock friction is provided by the empirical rate-and-state dependent friction relationship[12,13,16,17]. This framework considers the characteristic properties of fault surface including slip distance, slip velocity, normal stress, and state variables. However, this model does not include mechanisms such as fracturing during fault formation and evolution over tectonic time scales. Processes including fault damage, formation, and material properties evolution are accounted in LTT geodynamics models such as DynEarthSol2D[57] (DES2D) and DynEarthSol3D (DES3D is developed from DES2D).

DES3D is a robust, adaptive, multi-dimensional, finite element method. It solves the momentum and energy conservation equations in Lagrangian form using

unstructured meshes[57]. It uses the explicit FLAC algorithm[58] to solve for momentum balance. This approach solves the dynamic problem, $\nabla.\boldsymbol{\sigma}_{ij} + \rho_n g_n^z = \rho_n \mathbf{a}_n$, for quasi-static equilibrium, $\nabla.\boldsymbol{\sigma}_{ij} + \rho_n g_n^z = 0$ in LTT models. $i$ and $j$ correspond to element wise and $n$ to nodal wise quantities. Where $\rho$, $\boldsymbol{\sigma}$, $\boldsymbol{a}$, and $g^z$ are density, Cauchy stress tensor, nodal acceleration, and gravity acceleration in the z direction. To approach equilibrium, FLAC damps the inertial part of the momentum equation by solving $\nabla.\boldsymbol{\sigma}_{ij} + 0.8 \cdot \mathrm{sgn}(\mathbf{v}_n)\mathbf{f}_n + \rho_n g_n^z = \rho_n \mathbf{a}_n$. FLAC also uses a mass scaling technique to increase the time steps during the interseismic period by several orders of magnitude[57]. One explicit time step in FLAC solves the following system of equations:

$$\nabla.\boldsymbol{\sigma}_{ij} + \rho_n g_n^z = \mathbf{f}_{\mathrm{damped}}, \tag{3}$$

$$\mathbf{f}_{\mathrm{damped}} = \rho_{\mathrm{fict}}\mathbf{a}_n - 0.8 \cdot \mathrm{sgn}(\mathbf{v}_n)\|\mathbf{f}_n\|, \tag{4}$$

$$v_{\mathrm{elastic}} = \sqrt{K/\rho_{\mathrm{fict}}} = c \cdot v_{\mathrm{tectonic}}. \tag{5}$$

$K$ is the bulk modulus of the material, $\rho_{\mathrm{fict}}$ is a fictitious scaled density, and $c$ is an empirical constant in the order of $10^4 - 10^8$. By solving Eqs. (3–5) each time steps, we obtain new nodal velocities and displacements. We can then update the state of stresses and forces with a constitutive relationship (rheology). Time steps are calculated with $\Delta t = \frac{1}{2}\frac{\Delta l_{\min}}{c \cdot v_{\mathrm{tectonic}}}$, which satisfies the Courant–Friedrichs–Lewy condition explicit time scheme. $\Delta l_{\min}$ is the minimum element length (smallest grid size) of the whole mesh. Supplementary Figure 1 illustrates the flow of the FLAC algorithm.

**History dependent plasticity**. In order to simulate fault formation by inelastic strain accumulation and slip events we use a Mohr–Coulomb elastic plastic formulation that includes a yield stress, a plastic flow law, and a friction law (Eq. 1). A shear zone forms as strain localizes when elastic stresses exceed yield stress interseismically and can further accumulate when the dynamic friction $\mu_d$ changes as a function of velocity during a slip event.

For each time step, in a given element of the fault zone and in the frame of reference of the principal stresses, the strain increments are given by $\Delta\boldsymbol{\varepsilon}_1 = \frac{\mathbf{v}_1\Delta t}{L}$ and $\Delta\boldsymbol{\varepsilon}_3 = \frac{\mathbf{v}_3\Delta t}{H}$. $\mathbf{v}_1$ and $\mathbf{v}_3$ are the velocity in maximum and minimum stress direction. $L$ and $H$ are the length and the thickness of the element. During one explicit time step, we first update the state of stress by loading the element elastically:

$$\boldsymbol{\sigma}_{1,\mathrm{elastic}}^{n+1} = \boldsymbol{\sigma}_1^n + (\lambda + 2G)\Delta\boldsymbol{\varepsilon}_1 + \lambda\Delta\boldsymbol{\varepsilon}_3, \tag{6}$$

$$\boldsymbol{\sigma}_{3,\mathrm{elastic}}^{n+1} = \boldsymbol{\sigma}_3^n + (\lambda + 2G)\Delta\boldsymbol{\varepsilon}_3 + \lambda\Delta\boldsymbol{\varepsilon}_1, \tag{7}$$

where $\lambda$ and $G$ are Lame parameters. The element is in failure if the shear stress exceeds the Mohr–Coulomb yield criterion. If the state of stress is containing within the yield stress envelope the fault zone continues loading elastically following Eqs (6 and 7). If the elements in the fault zone are in failure, we apply plastic corrections, and the new state of stress is:

$$\boldsymbol{\sigma}_1^{n+1} = \boldsymbol{\sigma}_{1,\mathrm{elastic}}^{n+1} - \beta\left(E_1 - E_2 N_\psi\right), \tag{8}$$

$$\boldsymbol{\sigma}_3^{n+1} = \boldsymbol{\sigma}_{3,\mathrm{elastic}}^{n+1} - \beta\left(E_2 - E_1 N_\psi\right), \tag{9}$$

$\beta$ corresponds to the magnitude of plastic strain accumulation along the yield stress envelope[57], which

$$\beta = \frac{\boldsymbol{\sigma}_1^{n+1} - N_{\phi_d}\boldsymbol{\sigma}_3^{n+1} + 2C\sqrt{N_{\phi_d}}}{E_1 - E_2\left(N_{\phi_d} + N_\psi\right) + E_1 N_{\phi_d} N_\psi}. \tag{10}$$

where $\boldsymbol{\sigma}_1^n$ and $\boldsymbol{\sigma}_3^n$ are the corrected principal stresses. $E_1 = (K_s + 4/3G)$, $E_2 = (K_s - 2/3G)$ where $K_s$ is the bulk modulus and $G$ the shear modulus. $C$, $\varphi_d$, and $\psi$ are the cohesion, the dynamic friction angle, the dilation angle respectively. $N_{\phi_d} = \left(1 + \sin\phi_d\right)/\left(1 - \sin\phi_d\right)$ and $N_\psi = (1 + \sin\psi)/(1 - \sin\psi)$. $\beta$ carries all the deformation history and damps the effect of the direct change in friction. Equations 8 and 9 can be written as a differential equation carrying the deformation history of the fault zone over each time step:

$$\Delta\tau_{\max} = \beta G\left(1 + N_\psi\right) \tag{11}$$

The equations above show that the state of stresses is time dependent and carries strain history. As a result the shear stress has damage memory in a way akin to what is observed in laboratory experiments[47–49]. The state of stresses in our model is history dependent and replaces the equation for the state evolution in the rate and state formulation[12–14,16,17]. Supplementary Figure 2 illustrates this plastic formulation of rate-dependent friction.

**Implementations for slip events**. In order to simulate tectonic processes across time scales, depart from LTT DES3D, we apply adaptive time-stepping technique[50]. Our simulation uses a second order explicit time-stepping algorithm for integration. The time step scales inversely to the maximum slip velocity magnitude, $v_{\max}$, in the model, $\Delta t = \frac{1}{2}\frac{\Delta x_{\min}}{c\Delta v_{\max}}$. We use $c = 10^5$ in this paper. When the maximum slip velocity reaches the shear velocity ($\sqrt{G/\rho}$ ~4900 km s$^{-1}$), we use the minimum time step, $\Delta t_{\min} = \frac{1}{5}\frac{\Delta x_{\min}}{v_{\mathrm{shear}}}$. Therefore, in this study, $\Delta t = \min\left(\frac{1}{2}\frac{\Delta x_{\min}}{c\Delta v_{\max}}, \frac{1}{5}\frac{\Delta x_{\min}}{v_{\mathrm{shear}}}\right)$, and $\rho_{\mathrm{fict}} = \frac{K}{(\min(c \cdot v_{\max}, v_{\mathrm{shear}}))^2}$. In this case, we solve Eqs 3–5 quasi-dynamically and damp seismic wave radiation with time step ranging from millisecond to 10 days (Supplementary Figure 3a) to account for changes in velocity of more than 10 orders of magnitudes. The advantage of this technique is that we can cut off the slip velocity through the fictitious density to frequencies corresponding to the slip magnitude below that of a fully dynamic earthquake simulation. If we set $\Delta t_{\min} = \frac{1}{5}\frac{\Delta x_{\min}}{c\Delta v_{\mathrm{shear}}}$, which $\Delta t = \min\left(\frac{1}{2}\frac{\Delta x_{\min}}{c\Delta v_{\max}}, \frac{1}{5}\frac{\Delta x_{\min}}{v_{\mathrm{shear}}}\right)$ and $\rho_{\mathrm{fict}} = \frac{K}{(\min(c \cdot v_{\max}, v_{\mathrm{shear}}))^2}$, the peak slip velocity of our simulated earthquake can reach to shear wave speed (Supplementary Figure 3b).

**Code availability**. All relevant computer codes in this work are available from the authors upon reasonable request.

## Data availability
All relevant data in this work are available from the authors upon reasonable request.

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

## Acknowledgements

L.L.L. and X.T. were partially supported during this study by 3 NSF EAR awards, EAR-Geophysics 1547532, EAR-Tectonics 1524729, and EAR-Geophysics 1547532. The authors would like to thank Dr. Eunseo Choi and Dr. Eh Tan for help in the development of DynEarthSol3D (https://bitbucket.org/tan2/dynearthsol3d). The authors also thank Dr. Yajing Liu, Dr Eunseo Choi, and James Biemiller for providing extremely valuable comments during this study.

## Author contributions

X.T. performed experiments and analyzed the data, L.L.L. supervised the PhD student. X.T. and L.L.L. designed the numerical method and research path.

## Additional information

**Competing interests:** The authors declare no competing interests.

