## [Peer Review File · Nature Communications]

Reviewers' comments:

Reviewer #1 (Remarks to the Author):

A report on the paper by Xinyue Tong and Luc L. Lavier “Plastic Formulation of Rate and State Dependent Friction: Emergence of Slip Transient and Earthquakes” for Nature Communications, MS. no. NCOMMS-18-15541-T.

The paper deals with numerical modeling and experiments to simulate re-occurring tectonic earthquake sources and permanent plastic slip evolving within time periods of thousands years.

The paper claims to develop an approach that considers that aging as a result of the damage history in the fault zone (p.1), which seems, as far as this reviewer understands, the main (and interesting) novelty.

The dynamic (inertial) effects are not considered, so the seismic waves are not involved in the model. As such, the earthquakes themselves (as a tectonic event emitting seismic waves as a main attribute) are not directly modeled - which is perhaps not explicitly stated in this paper, although it is very common in geophysical modeling that seismic sources and seismic waves are not simultaneously involved in one model.

The model is not explicitly stated in this paper while it seems to rely substantially on Ref.37 where even some equations occur, although there is not much about damage/aging in Ref.37 but rather temperature is involved there, in contrast to the paper under review - so the relation is not much clear. But I understand that this is the way how computational geophysics is presented, without much ambitions to transfer the knowledge towards next generations (who will already not have chance for personal communications).

Altogether, taken into account the context of extremely short review time in such open-access journals and of the standards of publishing geophysical computational modeling and simulations, due to my opinion this article *can be published* in this journal after reflecting the following comments/suggestions.

There are several statements concerning energy in this paper. But it is not clear whether there is some energetics (in the sense of some at least a formal energy conservation when frictional dissipation is included) behind this model. Actually, the state-and-rate friction models are known to have difficulties at this point. Perhaps, some words about it might be worth.

p.1: “quasi-dynamic” is quite unusual - I would expect “quasi-static” (which is much often used - see also scholar.google).

p.2: in the first displayed formula for μ_d , only rate (velocity) occurs but not a state (aging/damage). Thus calling it “the empirical rate and state dependent friction law” is confusing. (I understand that it arises after substitution of the steady-state value of the state variable, as e.g. in Ref.29.) Besides, μ_d can become

negative for fast movements (if $b > a$) or slow movements (if $a < b$), which is certainly nonphysical, contradicting 2nd law of thermodynamics. Although it is often considered in geophysical papers, some discussion might be worth. And the slip velocity V is a vector in multidimensional situations (mentioned on p.4) or even can be a negative scalar in the one-dimensional situation (which is probably focus of this paper when speaking about “a single spring slider system” on p.2, although it is not much clear from fast reading).

p.2: Ref. 37 is about “DynEarthSol2D” code, not “DynEarthSol3D” as (confusingly) stated in the paper under review. Perhaps, also the later version developed by the second author and coworker at <https://bitbucket.org/tan2/dyneathsol3d> should be mentioned.

p.9-10: The references should be polished:

The digital object identifiers are not cited precisely E.g. Ref.19 doi:Artn B0830810.1029/2010jb008188 should be doi:10.1029/2010JB008188 and several other cases should be polished too.

“doi:Doi” in Refs.30, 38 39 41 etc. is awful.

“Siam” in Ref.34 should be capitalized (an abbreviation of a renown organization).

The names of journals oscillates: “Journal of Geophysical Research: Solid Earth” vs. “J Geophys Res-Sol Ea” in several references.

p.2,7: numbering of the displayed formulas would be nicer when flushed to the right margin (or even better omitted as they are not referred in the text).

Reviewer #2 (Remarks to the Author):

This manuscript entitled "A Plastic Formulation of Rate and State Dependent Friction: Emergence of Slip Transient and Earthquakes" by Tong and Lavier has proposed new formulations of the friction law, in which rate-dependent (velocity-weakening) constitutive law is implemented into their previous model calculating long-term plastic deformation. Authors suggest that this formulation shows a similar behavior as the rate and state dependent friction law. The rate and friction law is applicable only to the fault surface. On the other hand, the new formulation can be applicable to the finite shear zone, which is more adequate for the geologic fault. The simulated shear zone produces fast and slow deformation according to the value of the rate-dependent parameter $a-b$. Based on the scaling relations between several source parameters, authors insist that fast and slow deformation observed in numerical simulations are regular and slow earthquake, respectively. Recent studies suggest that the mixture of brittle and ductile deformation produces slow earthquake. This study follows a series of such studies, and will interest many readers in this field. However, I think following points should be addressed before its publication.

Major comments

1. References in the introductory part

Some references cited in the introductory part of this manuscript seem inappropriate. For example, other papers, such as Kanamori & Rivera (2006) would be more suitable for the reference #1 regarding the energy partitioning during the regular earthquake. In the second paragraph, models for slow earthquake are introduced. For models including "heat pressurization or dehydration reactions", Segall et al. (2010), Suzuki & Yamashita (2009), and Liu and Rice (2007) should be cited. Geologic studies of slow earthquake, such as Fagereng and Sibson (2010), Fagereng et al. (2014), and Ujiie et al. (2018), should be also discussed in the introductory part to emphasize the importance of viscoelastic deformation.

2. Comparison with rate and state friction law

Authors compare the frictional behavior of the new formulation with the rate and state friction law in Figure 2c and 2d by showing the time evolution of direct and evolving effect of frictional strength. However, authors compare results of different condition: Figure 2d is the velocity step test, whereas Figure 2c is spontaneous instability. Authors should compare results in the same condition. For example, authors will be able to conduct velocity step test by changing the loading velocity with time.

3. Strain localization

Authors show that deformation style changes with $a-b$ value from fast one to slow one. In Figure 3, slip velocity on the fault interface is shown against time for several examples. What about the strain localization during the slip event in the shear zone? Is strain localized on the fault surface in fast slip, and distributed in the entire shear zone in slow slip? It is worth discussing the spatial-temporal distribution of strain.

4. Analytical prediction of μ_c

Authors reported that slip behavior changes around $\mu = 0.0006$. Can authors derive analytical expressions of the critical value using input parameters? What controls the characteristic slip distance?

5. Methodology description

Because detailed methodology is provided in the previous paper and the supplementary figure, it was difficult to follow the algorithm of the numerical simulations. A figure showing a flowchart of calculations in each time step will help readers' understanding.

Minor comments

1. Parameter values

Please provide values of all input parameters, such as reference frictional coefficient and dilation angle.

2. Stress drop for slow earthquake

Authors reported that slow deformation has stress drop one order of magnitude smaller than that of fast deformation. I think stress drop of slow earthquake is about 30 kPa; two order of magnitude smaller than regular earthquake (e.g., Ide et al., 2007). It should be noted.

3. Figure 4

It is difficult to distinguish two kinds of dashed lines for fast EQ and slow EQ in Figure 4.

4. Scaling relations

In Figure 5, authors show the scaling relation between seismic moment and duration of events. Scaling relations for the constant fault size have steeper curves compared with the observed one. Are those data really consistent with the observed scaling relation? Recent papers (e.g., Ide & Maury, 2018) also report that there may be kink in the scaling relation of slow earthquake. Can you observe such features in the simulated slow deformation?

Reference

- Kanamori, Hiroo and Rivera, Luis (2006) Energy partitioning during an earthquake. In: Earthquakes: Radiated Energy and the Physics of Faulting. Geophysical Monograph Series. No.170. American Geophysical Union, Washington, DC, pp. 3-13. ISBN 978-0-87590-435-1
- Segall, P., A. M. Rubin, A. M. Bradley, and J. R. Rice (2010), Dilatant strengthening as a mechanism for slow slip events, *J. Geophys. Res.*, 115, B12305, doi:10.1029/2010JB007449.
- Segall, P., A. M. Rubin, A. M. Bradley, and J. R. Rice (2010), Dilatant strengthening as a mechanism for slow slip events, *J. Geophys. Res.*, 115, B12305, doi:10.1029/2010JB007449.
- Segall, P., A. M. Rubin, A. M. Bradley, and J. R. Rice (2010), Dilatant strengthening as a mechanism for slow slip events, *J. Geophys. Res.*, 115, B12305, doi:10.1029/2010JB007449.
- Suzuki, T., and T. Yamashita (2009), Dynamic modeling of slow earthquakes based on thermoporoelastic effects and inelastic generation of pores, *J. Geophys. Res.*, 114, B00A04, doi:10.1029/2008JB006042.
- Liu, Y., and J. R. Rice (2007), Spontaneous and triggered aseismic deformation transients in a subduction fault model, *J. Geophys. Res.*, 112, B09404, doi:10.1029/2007JB004930.
- Åke Fagereng, Richard H. Sibson; Mélange rheology and seismic style. *Geology*; 38 (8): 751–754.
- Fagereng, Å., G. W. B. Hillary, and J. F. A. Diener (2014), Brittle-viscous deformation, slow slip, and tremor, *Geophys. Res. Lett.*, 41, 4159–4167, doi:10.1002/2014GL060433.
- Ujiie, K., Saishu, H., Fagereng, Å., Nishiyama, N., Otsubo, M., Masuyama, H., & Kagi, H. (2018). An explanation of episodic tremor and slow slip constrained by crack-seal veins and viscous shear in subduction mélange. *Geophysical Research Letters*, 45.
- S. Ide, G. C. Beroza, D. R. Shelly, T. Uchide, *Nature* 447, 76 (2007).
- Ide, S., & Maury, J. (2018). Seismic moment, seismic energy, and source duration of slow earthquakes: Application of Brownian slow earthquake model to three major subduction zones. *Geophysical Research Letters*, 45, 3059–3067.

Author Response

A report on the paper by Xinyue Tong and Luc L. Lavier “Plastic Formulation of Rate and State Dependent Friction: Emergence of Slip Transient and Earthquakes” for Nature Communications, MS. no. NCOMMS-18-15541-T.

The paper deals with numerical modeling and experiments to simulate re-occurring tectonic earthquake sources and permanent plastic slip evolving within time periods of thousands years.

The paper claims to develop an approach that considers that aging as a result of the damage history in the fault zone (p.1), which seems, as far as this reviewer understands, the main (and interesting) novelty.

The dynamic (inertial) effects are not considered, so the seismic waves are not involved in the model. As such, the earthquakes themselves (as a tectonic even emitting seismic waves as a main attribute) are not directly modeled - which is perhaps not explicitly stated in this paper, although it is very common in geophysical modeling that seismic sources and seismic waves are not simultaneously involved in one model.

Authors' response: Thanks to the reviewer for pointing this out. The explicit Fast Lagrangian Analysis of Continua (FLAC) algorithm has been further developed for studies in Ref. 37 (it is now Ref.57 in the revised manuscript but we will use Ref.37 in this response for consistence) and this manuscript to solve for momentum balance. While FLAC damps the momentum equation for equilibrium with an inertial component resolving Earthquakes slip velocities, the spectrum corresponding to emitted seismic waves is not resolved in this study. It could be done with a dynamic time step small enough to resolve elastic wave propagation. Our study in this manuscript approximately account for inertial effect but not seismic radiation (through a radiation damping term). Therefore, our approach is partially quasi dynamic, as quasi static means that no inertial component is resolved. We have changed the word “quasi-dynamic” to “damped quasi-dynamic” in our manuscript.

The model is not explicitly stated in this paper while it seems to rely substantially on Ref.37 where even some equations occur, although there is not much about damage/aging in Ref.37 but rather temperature is involved there, in contrast to the paper under review - so the relation is not much clear. But I understand that this is the way how computational geophysics is presented, without much ambitions to transfer the knowledge towards next generations (who will already not have chance for personal communications).

We thank the author for his concerns. DES3D is available freely to the community through bitbucket (<https://bitbucket.org/tan2/dynearthsol3d>) and all changes made through this publication will be made available as soon as the paper is published. As the reviewer commented, the Ref. 37 is about “DynEarthSol2D” (DES2D), which is a code aimed to succeed the geoFLAC code¹. DES2D, written in Fortran, uses triangle meshes with adaptive mesh refinement technique which make the code as good as geoFLAC code but runs much faster (as it shown in Ref.37). After that, co-authors of Rep.37 have make efforts to develop “DynEarthSol3D” (DES3D). DES3D is written in C++, and it can model

deformations both in 2D and 3D. DES3D is in its final develop phase and a manuscript/document will be available in the near future. The study in this manuscript use DES3D with further implementation of damage/aging and adaptive time stepping technique.

Altogether, taken into account the context of extremely short review time in such open-access journals and of the standards of publishing geophysical computational modeling and simulations, due to my opinion this article can be published in this journal after reflecting the following comments/suggestions.

Authors' response: Thanks for the reviewer's recommendation. We have revised our manuscript according to the following comments/suggestions.

There are several statements concerning energy in this paper. But it is not clear whether there is some energetics (in the sense of some at least a formal energy conservation when frictional dissipation is included) behind this model. Actually, the state-and-rate friction models are known to have difficulties at this point. Perhaps, some words about it might be worth.

Authors' response: The statements concerning energy in this manuscript is really about energy partition between kinetic energy which the fault used to slip and strain energy which been used to damage the fault zone materials. This study shows that fast slips and slow slips differ in the way they partition energy.

We are aware that the rate-and-state friction models has difficulties in the framework of thermodynamics when $a-b > 0$. However, our study is not affected by these issues and represent a true physical model since, we only use the rate dependent friction law to calculate the friction coefficient (values have been provided in the response below) for Mohr-Coulomb criterion. In addition, our model results also match the instability analysis which is independent from the rate-and-state friction law.

p.1: "quasi-dynamic" is quite unusual - I would expect "quasi-static" (which is much often used - see also scholar.google).

Authors' response: We have change "quasi-dynamic" to "damped quasi-dynamic" in our manuscript.

p.2: in the first displayed formula for μ_d , only rate (velocity) occurs but not a state (aging/damage). Thus calling it "the empirical rate and state dependent friction law" is confusing. (I understand that it arises after substitution of the steady-state value of the state variable, as e.g. in Ref.29.) Besides, μ_d can become negative for fast movements (if $b > a$) or slow movements (if $a < b$), which is certainly nonphysical, contradicting 2nd law of thermodynamics. Although it is often considered in geophysical papers, some discussion might be worth. And the slip velocity V is a vector in multidimensional situations (mentioned on p.4) or even can be a negative scalar in the one-dimensional

situation (which is probably focus of this paper when speaking about “a single spring slider system” on p.2, although it is not much clear from fast reading).

Authors’ response: We thank the reviewer to help clarify our terminologies and wordings. We have changed the wording from “the empirical rate and state dependent friction law at steady state” to “the rate dependent friction law at steady state” in our manuscript.

In our simulations, the slip velocity on each nodes is a vector, but the velocity (magnitude) used to calculate friction coefficient is a scalar. We have change “V is slip velocity” to “V is velocity magnitude” in p.2 to clarify.

We are aware of the critical discussions on the limitations of rate-and-state friction law especially the arguments of its irrational thermodynamics². However, the concern doesn’t affect our study in this manuscript. In this study, we use $\mu_0 = 0.6$, $V_o = 10^{-6}$, $a - b = [-0.004, 0]$, and $V = [10^{-10}, 10^{-1}]$. According to $\mu_d = \left(\mu_0 + (a - b)\ln\frac{V}{V_o}\right)$, μ_d only varies within a small range from 0.55 to 0.64 in our experiments.

p.2: Ref. 37 is about “DynEarthSol2D” code, not “DynEarthSol3D” as (confusingly) stated in the paper under review. Perhaps, also the later version developed by the second author and coworker at <https://bitbucket.org/tan2/dyneearthsol3d> should be mentioned.

Authors’ response: The relationship between DynEarthSol2D and DynEarthSol3D has been cleared in the authors’ response above. For clarification, we add “(DES3D, it is developed from DynEarthSol2D)” in the third paragraph of Method section. The co-developers of DES3D will be mentioned in the End Notes section.

p.9-10: The references should be polished:

The digital object identifiers are not cited precisely E.g. Ref.19 doi:Artn Bo830810.1029/2010jbo08188 should be doi:10.1029/2010JB008188 and several other cases should be polished too.

“doi:Doi” in Refs.30, 38 39 41 etc. is awful.

“Siam” in Ref.34 should be capitalized (an abbreviation of a renown organization).

The names of journals oscillates: “Journal of Geophysical Research: Solid Earth” vs. “J Geophys Res-Sol Ea” in several references.

Authors’ response: Thanks for reviewer to point those mistakes out. We have polished the references to meet the Nature Communications format style.

p.2,7: numbering of the displayed formulas would be nicer when flushed to the right margin (or even better omitted as they are not referred in the text).

Authors’ response: We have flushed them to the right margin.

References:

- 1 Cundall, P. A. Numerical Experiments on Localization in Frictional Materials. *Ing Arch* **59**, 148-159, doi:10.1007/Bf00538368 (1989).
- 2 Roubicek, T. A note about the rate-and-state-dependent friction model in a thermodynamic framework of the Biot-type equation. *Geophys J Int* **199**, 286-295, doi:10.1093/gji/ggu248 (2014).

Reviewer #2 (Remarks to the Author):

This manuscript entitled “A Plastic Formulation of Rate and State Dependent Friction: Emergence of Slip Transient and Earthquakes” by Tong and Lavier has proposed new formulations of the friction law, in which rate-dependent (velocity-weakening) constitutive law is implemented into their previous model calculating long-term plastic deformation. Authors suggest that this formulation shows a similar behavior as the rate and state dependent friction law. The rate and friction law is applicable only to the fault surface. On the other hand, the new formulation can be applicable to the finite shear zone, which is more adequate for the geologic fault. The simulated shear zone produces fast and slow deformation according to the value of the rate-dependent parameter $a-b$. Based on the scaling relations between several source parameters, authors insist that fast and slow deformation observed in numerical simulations are regular and slow earthquake, respectively. Recent studies suggest that the mixture of brittle and ductile deformation produces slow earthquake. This study follows a series of such studies, and will interest many readers in this field. However, I think following points should be addressed before its publication.

Authors' response: Thanks for the reviewer's recommendation. We have revise our manuscript to reflect the following comments.

Major comments

1. References in the introductory part

Some references cited in the introductory part of this manuscript seem inappropriate. For example, other papers, such as Kanamori & Rivera (2006) would be more suitable for the reference #1 regarding the energy partitioning during the regular earthquake. In the second paragraph, models for slow earthquake are introduced. For models including “heat pressurization or dehydration reactions”, Segall et al. (2010), Suzuki & Yamashita (2009), and Liu and Rice (2007) should be cited. Geologic studies of slow earthquake, such as Fagereng and Sibson (2010), Fagereng et al. (2014), and Ujiie et al. (2018), should be also discussed in the introductory part to emphasize the importance of viscoelastic deformation.

Authors' response: We used Kanamori & Rivera (2006) to replace reference #1. We added Segall et al., (2010), Suzuki & Yamashita (2009), and Liu and Rice (2007) as references for “heat and pressurization or dehydration reactions”. We added Fagereng and Sibson (2010), Fagereng et al. (2014), and Ujiie et al. (2018) as references for the viscoelastic deformation statements in this manuscript. In order to keep the coherence, we didn't add more sentences but we certainly emphasize the importance of viscoelastic deformation as it is “more adequate for the pressure and temperature conditions under which deep SSEs are observed”. In fact, the second co-author of this manuscript is among the first to discuss the viscoelastic deformation and transient events both in numerical study (Lavier et al., 2013) and field observation (Hayman and Lavier, 2014).

2. Comparison with rate and state friction law

Authors compare the frictional behavior of the new formulation with the rate and state friction law in Figure 2c and 2d by showing the time evolution of direct and evolving effect of frictional strength. However, authors compare results of different condition: Figure 2d is the velocity step test, whereas Figure 2c is spontaneous instability. Authors should compare results in the same

condition. For example, authors will be able to conduct velocity step test by changing the loading velocity with time.

Authors' response: Thanks for the reviewer's critical comment. We have use our model to setup a velocity stepping test in tectonic scale. We redraw the figure 2 and rewrite the caption and associated text in the manuscript.

3. Strain localization

Authors show that deformation style changes with a-b value from fast one to slow one. In Figure 3, slip velocity on the fault interface is shown against time for several examples. What about the strain localization during the slip event in the shear zone? Is strain localized on the fault surface in fast slip, and distributed in the entire shear zone in slow slip? It is worth discussing the spatial-temporal distribution of strain.

Authors' response: Strain do localize on the fault surface for both fast slip and slow slip. The strain location has a finite thickness and it is independent of the different slip behaviors. the thickness of the localized strained zone is dependent on mesh size. We have a section in supplementary materials discusses the mesh resolution dependence of fault zone evolution.

4. Analytical prediction of μ_c

Authors reported that slip behavior changes around $\mu = 0.0006$. Can authors derive analytical expressions of the critical value using input parameters? What controls the characteristic slip distance?

Authors' response: We can analytically derive the critical value of μ using input parameters and a characteristic slip distance. However, the values of characteristic slip distance from published studies are not consistent. There are orders of magnitude difference between lab experiments derived values and field observed values from earthquakes. Therefore, our approach is to treat the characteristic slip distance as an unknown. We use the critical value of $\Delta\mu_c = 0.0006$ and other input parameters to derive the characteristic slip distance. The detailed analysis is shown in Paragraph 6 of the main text. For our model with a fault length of 60 km, the characteristic slip distance is about 0.15 m. Other than the critical value on friction change, the characteristic slip distance is also depends on the shear modulus, normal stress, and fault zone length.

5. Methodology description

Because detailed methodology is provided in the previous paper and the supplementary figure, it was difficult to follow the algorithm of the numerical simulations. A figure showing a flowchart of calculations in each time step will help readers' understanding.

Authors' response: To help readers' understanding, we add two figures in supplementary material. Supplementary figure 1 shows a flowchart of the FLAC algorithm in each time step. Supplementary figure 2 shows the implication of rate-dependent friction in Mohr-Coulomb Failure Criterion for elasto-plastic rheology.

Minor comments

1. Parameter values

Please provide values of all input parameters, such as reference frictional coefficient and dilation angle.

Authors' response: We add a table which provides values of all input parameters in supplementary table 1.

2. Stress drop for slow earthquake

Authors reported that slow deformation has stress drop one order of magnitude smaller than that of fast deformation. I think stress drop of slow earthquake is about 30 kPa; two order of magnitude smaller than regular earthquake (e.g., Ide et al., 2007). It should be noted.

Authors' response: As we mentioned in paragraph 10, the stress drop that presented in this study is direct measurements of maximum shear stress changes before and after a slip event. However, the stress drop from fast and slow earthquakes are derived values. In addition, our model setup is not representing any realistic geologic structures but a single spring-slider system in tectonic scale. Therefore, we don't compare exact values and magnitudes between modeled results and observations. We are focusing on the scaling relationships. Specifically, for stress drop, as shown in figure 4a., the ranges for fast and slow earthquakes are 10^1 - 10^{-1} MPa and 10^0 - 10^{-2} MPa, which we stated that the stress drop for slow slip is about one order of magnitude smaller than that of fast deformation. We realized that two order of difference has been observed (e.g., Ide et al., 2007), but the point we try to make here is that the scaling is right. Observations (e.g. Ide et al., 2007) has shown that, for both fast and slow slips, stress drop is linearly scales with strain drop (co-seismic slip divided by fault length). Our modeled results (Figure 4a) also show this linearly relationship (statistically).

3. Figure 4

It is difficult to distinguish two kinds of dashed lines for fast EQ and slow EQ in Figure 4.

Authors' response: Following the response above, the two red dashed lines in figure 4a have same slope which most slip events are plotted inside those two lines. It means that both simulated fast and slow slips are statistically following the linearly relationship between stress drop and strain drop.

The red dashed line in figure 4b shows the linearly scaling relationship between co-seismic slip and duration for fast slips. The blue dashed line shows a different scaling for slow slip events.

4. Scaling relations

In Figure 5, authors show the scaling relation between seismic moment and duration of events. Scaling relations for the constant fault size have steeper curves compared with the observed one. Are those data really consistent with the observed scaling relation? Recent papers (e.g., Ide & Maury, 2018) also report that there may be kink in the scaling relation of slow earthquake. Can you observe such features in the simulated slow deformation?

Authors' response: In figure 5, the scaling relations for constant fault size have steeper slopes. These steeper slopes are originating from the coseismic slip and duration scaling shown in figure 4b. We use co-seismic slip to calculate Moment. Therefore, for fast slips, it shows a $M_0 \sim T$ scaling for constant fault size; for slow slips, it shows a $M_0 \sim T^{1/15}$ scaling for constant fault size. Statistically, our modeled results consistent with observed scaling relations for both fast and slow deformations.

The slip area of the Brownian Slow Earthquake (BSE) model (of 2D surface) shown in Ide & Maury, 2018 can change over each time step. The kinks in the scaling relation of slow earthquakes between two different input characteristic times are in a range which is smaller than the study in this manuscript. Our model (of 2D cross-section) setup is very simple and mimic a tectonic-scale spring-slider system, and the slip area is not changing over time for each

experiments. This simplified model setup would not simulate such features, and we are not aiming to introduce such complexity into this manuscript. However, it is worth to point out that the scaling of BSE model turns to a linear relation at the region of larger moments.

Reference

- Kanamori, Hiroo and Rivera, Luis (2006) Energy partitioning during an earthquake. In: Earthquakes: Radiated Energy and the Physics of Faulting. Geophysical Monograph Series. No.170. American Geophysical Union, Washington, DC, pp. 3-13. ISBN 978-0-87590-435-1
- Segall, P., A. M. Rubin, A. M. Bradley, and J. R. Rice (2010), Dilatant strengthening as a mechanism for slow slip events, *J. Geophys. Res.*, 115, B12305, doi:10.1029/2010JB007449.
- Segall, P., A. M. Rubin, A. M. Bradley, and J. R. Rice (2010), Dilatant strengthening as a mechanism for slow slip events, *J. Geophys. Res.*, 115, B12305, doi:10.1029/2010JB007449.
- Segall, P., A. M. Rubin, A. M. Bradley, and J. R. Rice (2010), Dilatant strengthening as a mechanism for slow slip events, *J. Geophys. Res.*, 115, B12305, doi:10.1029/2010JB007449.
- Suzuki, T., and T. Yamashita (2009), Dynamic modeling of slow earthquakes based on thermoporoelastic effects and inelastic generation of pores, *J. Geophys. Res.*, 114, B00A04, doi:10.1029/2008JB006042.
- Liu, Y., and J. R. Rice (2007), Spontaneous and triggered aseismic deformation transients in a subduction fault model, *J. Geophys. Res.*, 112, B09404, doi:10.1029/2007JB004930.
- Åke Fagereng, Richard H. Sibson; Mélange rheology and seismic style. *Geology*; 38 (8): 751–754.
- Fagereng, Å., G. W. B. Hillary, and J. F. A. Diener (2014), Brittle-viscous deformation, slow slip, and tremor, *Geophys. Res. Lett.*, 41, 4159–4167, doi:10.1002/2014GL060433. ^[SEP]
- Ujiie, K., Saishu, H., Fagereng, Å., Nishiyama, N., Otsubo, M., Masuyama, H., & Kagi, H. (2018). An explanation of episodic tremor and slow slip constrained by crack-seal veins and viscous shear in subduction mélange. *Geophysical Research Letters*, 45.
- S. Ide, G. C. Beroza, D. R. Shelly, T. Uchide, *Nature* 447, 76 (2007).
- Ide, S., & Maury, J. (2018). Seismic moment, seismic energy, and source duration of slow earthquakes: Application of Brownian slow earthquake model to three major subduction zones. *Geophysical Research Letters*, 45, 3059–3067.

Reference:

- Hayman, N. W., and Lavier, L. L., 2014, The geologic record of deep episodic tremor and slip: *Geology*, v. 42, no. 3, p. 195-198.
- Lavier, L. L., Bennett, R. A., and Duddu, R., 2013, Creep events at the brittle ductile transition: *Geochemistry Geophysics Geosystems*, v. 14, no. 9, p. 3334-3351.